# ON MONOTONICITY IN AI ALIGNMENT

## ABSTRACT

Comparison-based preference learning has become central to the alignment of AI models with human preferences. However, these methods may behave counterintuitively. After empirically observing that, when accounting for a preference for response $y$ over $z$, the model may actually decrease the probability (and reward) of generating $y$, (an observation also made by others), this paper investigates the root causes of (non) monotonicity. We first propose a framework for general comparison-based preference learning that subsumes Direct Preference Optimization (DPO), Generalized Preference Optimization (GPO) and Generalized Bradley-Terry (GBT). We prove that, under mild assumptions, such methods guarantee that the score difference between the chosen and rejected alternative increases, which we call *pairwise monotonicity*. We also provide necessary and sufficient conditions for increase of the score of the chosen (rejected) alternative, which we call *individual monotonicity*. Notably, our theory shows that some flavors of individual monotonicity are too demanding in practice. These results clarify the limitations of current methods, and provide guidance for developing more trustworthy preference learning algorithms.

## 1 INTRODUCTION

Large AI models and large language models (LLMs) in particular now power an ever-growing range of user-facing applications, from conversational assistants to code-completion systems, and their societal impact expands with every deployment. Ensuring that these models behave in accordance with human preferences has therefore become a defining challenge. Comparison-based preference learning, in which annotators rank or choose among candidate outputs and the model is fine-tuned to reproduce those choices, has emerged as the workhorse paradigm for alignment. Although simple to describe and remarkably effective in practice, this paradigm conceals subtle theoretical pitfalls that undermine our ability to reason about, and ultimately trust, the models it produces.

The most widely used framework for comparison-based preference learning is Reinforcement Learning from Human Feedback (RLHF) (Christiano et al., 2017; Stiennon et al., 2020), which in practice often reduces to Direct Preference Optimization (DPO) (Rafailov et al., 2023) or its recent generalizations (Tang et al., 2024; Azar et al., 2024; Fageot et al., 2024). The core intuition behind these methods is straightforward: if a human prefers response $y$ over response $z$, the fine-tuned model should boost the likelihood of $y$ and suppress that of $z$. However, perhaps surprisingly, recent empirical work has shown that this intuition can fail in practice. In some cases, fine-tuning on a preference pair where $y$ beats $z$ actually reduces the model's probability or logit score for $y$ (Pal et al., 2024; Razin et al., 2024). Such counterintuitive properties raise serious concerns: they erode trust in the training procedure, complicate the design of data-collection protocols, and may even incentivize annotators to misreport their true preferences, in high-stakes applications. These phenomena call for a fundamental question:

*What monotonicity guarantees do comparison-based preference learning algorithms provide?*

In this paper, we provide the first systematic study of monotonicity for a broad class of comparison-based preference learning methods, which includes DPO, Generalized Preference Optimization (GPO), and Generalized Bradley-Terry (GBT). Specifically, our contributions are:

- We formalize a rich variety of flavors of *monotonicity*, structured around various considerations (pairwise/individual, local/global, score/probability, minimum/gradient-descent).

- We prove that, a general comparison-based preference learning framework, which includes DPO, GPO and GBT, guarantees *local pairwise monotonicity*.
- We identify sufficient conditions for, global pairwise, local individual-score, local individual-probability gradient-descent pairwise, gradient-descent individual-score and gradient-descent individual-probability monotonocity.

The paper is organized as follows. Section 2 reviews related work, and exhibits an empirical setting where monotonicity fails. Section 3 introduces a general comparison-based preference learning framework that encompasses most leading solutions. Section 4 presents our main result, on *local pairwise monotonicity*. Section 5 discusses other forms of monotonicity. Section 6 concludes. Table 1 summarizes the notation frequently used.

## 2 CONTEXT AND MOTIVATIONS

**The Bradley-Terry model and its generalizations.**    Comparison-based preference learning builds upon a large literature, which started with the seminal works of Thurstone (1927), Zermelo (1929), and then Bradley & Terry (1952). Their solution relies on a probabilistic model of how some ground-truth preference gets distorted into reported comparative judgments, thereby enabling preference learning from inconsistent data. Their model was later generalized by Luce (1959) and Plackett (1975) to account for the selection of one preferred alternative out of many, by Kristof et al. (2019) and Fageot et al. (2024) to enable quantified comparative judgments, and by Menke & Martinez (2008), Guo et al. (2018), Noothigattu et al. (2018), Lee et al. (2019) and Blanchard et al. (2025) to learn linear models of preferences, and thus generalize beyond the specific compared items.

**Nonlinear models with a Bradley-Terry loss.**    Csiszár (2012) and Zhao et al. (2016) are some of the earliest nonlinear models whose loss functions are constructed based on comparative judgments and on the Bradley-Terry loss. More recently, with the rise of language models (Vaswani et al., 2017; Brown et al., 2020) and of the alignment problem (Hadfield-Menell & Hadfield, 2019; Hoang, 2019), the Bradley-Terry loss was proposed to fine-tune language models to reported comparative human judgments, e.g. through the convoluted *Reinforcement Learning with Human Feedback* (RLHF) (Christiano et al., 2017; Stiennon et al., 2020). This approach was later shown to be reducible to *Direct Preference Optimization* (DPO) (Rafailov et al., 2023), where model fine-tuning boils down to minimizing a Bradley-Terry-derived loss function of the language model parameters. Lately, alternative loss functions were proposed, which typically replace the Bradley-Terry loss with an alternative term (Tang et al., 2024; Azar et al., 2024). The global preference-learning framework has also been used for other use cases, like image captioning (L et al., 2024) and policy tuning (Hejna et al., 2023), as well as image (Liang et al., 2024; Liu et al., 2024a) sound (Zhang et al., 2024) and video generation (Dai et al., 2024).

**Monotonicity.**    While RLHF and DPO have by now been widely used to align language models, little is known about their actual mathematical guarantees. For instance, recently, Chen et al. (2024) pointed out that order often failed to be recovered by preference learning algorithm. More strikingly, Pal et al. (2024); Razin et al. (2024) made observations akin to ours, as they also witness a decrease

Table 1: Summary of notations

| Symbol | Set | Meaning |
|---|---|---|
| $x$ | $\mathcal{B}$ | Background for elements to be scored (eg prompts) |
| $y, z$ | $\mathcal{A}$ | Items to be scored (e.g., responses to prompt) |
| $c$ | $\mathcal{C}$ | quantitative comparison value (between two items $y, z$) |
| $(x, y, z, c)$ | $\mathbf{D}$ | One data point of the dataset $\mathbf{D}$ |
| $\theta$ | $\mathbb{R}^d$ | Parameter of a model |
| $s_{y\|x}(\theta)$ | $\mathbb{R}$ | Score (*logits*) of model $\theta$ to item $y$ in background $x$ |
| $s_{yz\|x}(\theta)$ | $\mathbb{R}$ | Score difference of model $\theta$ to item $y$ and $z$ in background $x$ ($= s_{y\|x}(\theta) - s_{z\|x}(\theta)$) |
| $\mathcal{R}(\theta)$ | | Regularizer function |
| $\ell(s, c)$ | | Point-wise loss function between score $s$ and comparison $c$ |
| $\pi_\theta(y\|x)$ | $[0, 1]$ | Probability of response $y$ in background $x$ for model $\theta$ |
| $f$ | | A probability density over $\mathcal{C}$, the GBT "root law" |
| $\Phi_f(s)$ | | The cumulant-generating function of root law $f$ ($= \log \int_{\mathcal{C}} e^{s\gamma} f(\gamma) d\gamma$) |

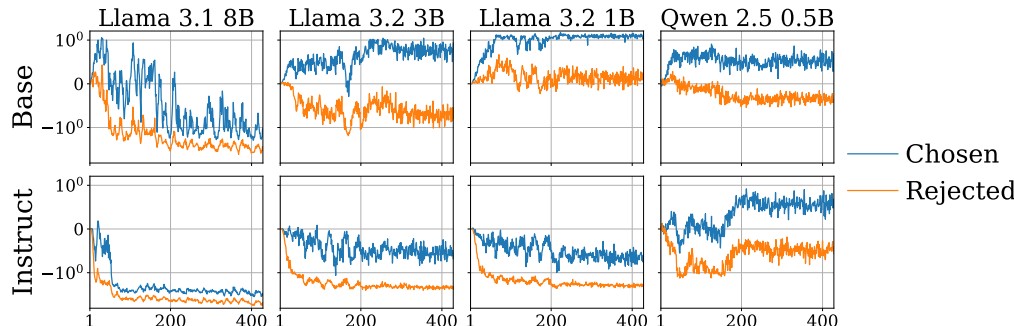

Figure 1: Evolution of the average score of chosen and rejected alternatives on the validation set, over the DPO training of several LLMs for one epoch. One could expect the chosen (rejected) response curves to be above (below) zero. This is not the case.

of the probability of the preferred alternative, after including the comparison that says that it is preferred in gradient descent. In fact, there is a growing literature on fixes to the DPO loss (Pang et al., 2024; Liu et al., 2024b). Conversely, Noothigattu et al. (2020) proves fully-pairwise monotonicity (Definition 7) for certain non-generalizing extensions of the Bradley-Terry model. The result was then generalized by Fageot et al. (2024) to (non-generalizing) Generalized Bradley-Terry (GBT), and then by Blanchard et al. (2025) to a subclass of linear (generalizing) GBT models. As opposed to the negative experimental findings, these three positive results focus on the monotonicity of the loss minimum, upon the addition or modification of a reported comparative judgment. We believe this to yield a complementary, and perhaps more fundamental, insight than the study of gradient descent.

**Motivating example.** We report in Figure 1, for several LLMs, the scores of the chosen (resp. rejected) alternative for the trained model, relative to the score of the reference model $\theta^{\text{ref}} = \theta_0$. Namely, we report on the first and second row:

$$\Delta s_{\text{chosen}}(t) = s_{\text{chosen}}(\theta_t) - s_{\text{chosen}}(\theta^{\text{ref}}) \qquad \Delta s_{\text{rej.}}(t) = s_{\text{rej.}}(\theta_t) - s_{\text{rej.}}(\theta^{\text{ref}}),$$

where $\theta_t$ denotes the model parameters at iteration $t$, $\theta^{\text{ref}} = \theta_0$ denotes the reference model, and $s_{\text{chosen}}$ ($s_{\text{rej.}}$) denote the mean of the scores of the chosen (rejected) alternatives on the iteration batch.

A first observation is that the chosen curve is always above the rejected curve $\Delta s_{\text{chosen}} > \Delta s_{\text{rej.}}$. Equivalently, there holds $s_{\text{chosen}}(\theta_t) - s_{\text{rej.}}(\theta_t) > s_{\text{chosen}}(\theta^{\text{ref}}) - s_{\text{rej.}}(\theta^{\text{ref}})$: the so-called *margin* is increased during the training. We study the behavior of the margin in the forthcoming section 4 under the name "pairwise monotonicity". This observation is consistent with the fact that we use RLHF and DPO, which are designed to increase the margin.

A second, more puzzling, observation is that the chosen curves are negative during a portion of the training for certain Base models, and all Instruct models. The rejected curve is also positive in certain cases (Llama 3.2 1B Base). As each datapoint is only seen once during the training, the finetuning effectively *decreases* the scores of the preferred alternatives, and at times increases the scores of the rejected alternatives. We study the evolution of the chosen (and rejected) scores upon an optimization step in the forthcoming section 5, under the name "individual monotonicity".

We study 6 Llama models (3.1 8B, 3.2 3B, 3.2 1B) and one Qwen model (2.5 0.5B) (all *base* and *instruct* variants) (AI@Meta, 2024; Qwen et al., 2025) and UltraFeedback (Cui et al., 2024). We used torchtune (torchtune maintainers & contributors, 2024) with a modified "full_dpo_distributed" recipe (provided in the Supplementary Material). The training consists of one epoch on the dataset, with batchsize 128. Our experiments ran on a compute node of 8 H100, for less than 100 GPU-hours.

## 3    A GENERAL COMPARISON-BASED PREFERENCE LEARNING FRAMEWORK

In this section, we introduce a general comparison-based preference learning framework, that encompasses most leading methods, including Bradley-Terry (BT), Generalized Bradley-Terry (GBT), Direct Preference Optimization (DPO), and General Preference Optimization (GPO).

Consider a set $\mathcal{A}$ of alternatives to be scored. We assume that their scoring is dependent on a background $\mathcal{B}$. Typically, in the context of language model alignment, $\mathcal{B}$ would be the set of prompts and $\mathcal{A}$ would be the set of responses to the prompt. Denote $s : \mathcal{A} \times \mathcal{B} \times \mathbb{R}^D \to \mathbb{R}$ the parameterized scoring function to be learned, where $s_{y|x}(\theta) \in \mathbb{R}$ is the score assigned to alternative $y \in \mathcal{A}$ given background $x \in \mathcal{B}$ for a parameter vector $\theta \in \mathbb{R}^D$.

The parameter vector $\theta$ is typically learned by fitting a comparison-based preference multiset $\mathbf{D} \triangleq (\mathcal{B} \times \mathcal{A} \times \mathcal{A} \times \mathcal{C})^*$ composed of a finite number of conditional pairwise response comparisons $(x, y, z, c)$, where $x \in \mathcal{B}$ is the background (e.g. prompt), $y, z \in \mathcal{A}$ are proposed alternatives (e.g. responses) to $x$, and $c \in \mathcal{C} \subset \mathbb{R}$ says whether $y$ was preferred over $z$ ($c > 0$), or $z$ was preferred over $y$ ($c < 0$). Typically, assuming binary comparisons, we would have $\mathcal{C} \triangleq \{-1, +1\}$, with $c = 1$ if $y$ was preferred to $z$, and $c = -1$ otherwise.

To fit $\theta$ to $\mathbf{D}$, we assume that a loss is minimized. Denoting $s_{yz|x}(\theta) \triangleq s_{y|x}(\theta) - s_{z|x}(\theta)$ the score difference between responses $y$ and $z$ on prompt $x$, we consider the following general loss form:

$$\text{Loss}(\theta|\mathbf{D}) = \mathcal{R}(\theta) + \sum_{(x,y,z,c)\in\mathbf{D}} \ell(s_{yz|x}(\theta), c),$$

where $\mathcal{R} : \mathbb{R}^D \to \mathbb{R}$ is a (potentially nil) regularization and $\ell : \mathbb{R} \times \mathcal{C} \to \mathbb{R}$ is the loss per data point.

In the sequel, we show that our setting generalizes most state-of-the-art solutions for comparison-based preference learning, which are obtained by instantiating different scoring functions $s$ and different per-data losses $\ell$. We note that some models escape our formalism, as their losses also depend on $s_{y|x}(\theta)$ or $\pi_\theta(y|x)$; see e.g. Pal et al. (2024); Xiao et al. (2024); Meng et al. (2024).

## 3.1 VARIANTS OF THE SCORING FUNCTION $s$

**Linear model.** Common scoring functions $s$ in machine learning rely on linear models. To do so, consider a fixed embedding map $f : \mathcal{B} \times \mathcal{A} \to \mathbb{R}^D$, and a score function $s_{y|x}(\theta) = \theta^\top f(x, y)$. This is, to a certain extent, what is performed in Reinforcement Learning with Human Feedback (RLHF), where the score (also known as reward) is constructed as a linear function of an embedding. Note that this is only one step of RLHF, which also involves policy optimization given a scoring function.

**Language models.** For language models, we have $\mathcal{A} = \mathcal{B} = \mathbf{A}^* \triangleq \bigcup_{n\in\mathbb{N}} \mathbf{A}^n$, i.e. both the alternatives and the background are finite sequences of characters of a finite alphabet $\mathbf{A}$. The scoring function then assigns a score $s_{y|x}(\theta) \in \mathbb{R}$ to any response (alternative) $y \in \mathcal{A}$ under a prompt (background) $x \in \mathcal{B}$. It typically corresponds to the last layer of the language model, before a softmax operator is applied to derive a probability distribution over $\mathcal{A}$, i.e. it is common to set

$$\pi_\theta(y|x) \triangleq \frac{\exp(s_{y|x}(\theta))}{\sum_{z\in\mathbf{A}^*} \exp(s_{z|x}(\theta))},$$

where $\pi_\theta(y|x)$ is the probability of response $y$ under prompt $x$. If so, the scores $s_{y|x}(\theta)$ are known as the *logits* of the generative model.

**Direct Preference Optimization (DPO).** In Direct Preference Optimization (DPO), which is an equivalent more direct reformulation of RLHF, a reference model $\pi_{ref} : \mathbf{A}^* \to \Delta(\mathbf{A}^*)$ is used to bound the variations of the scores. The score $s_{y|x}(\theta)$ to response $y$ conditionally to prompt $x$ assuming model $\theta$ is then given by

$$s_{y|x}(\theta) = \beta \log \frac{\pi_\theta(y|x)}{\pi_{ref}(y|x)} + \beta \log Z_x(\theta),$$

where $Z_x(\theta) = \sum_y \pi_{ref}(y|x) \exp(\beta^{-1} s_{y|x}(\theta))$ is the partition function of $\pi_\theta(\cdot|x)$, and $\beta \in \mathbb{R}_{\geq 0}$ is a positive scalar hyperparameter. Note that $s_{y|x}(\theta)$ is here often known as the *reward*.

In all these cases, $s_{y|x}$ is often assumed to be differentiable, if not smooth[1]. In the sequel, we will assume that it is continuously differentiable.

**Assumption 1.** *For all $x \in \mathcal{B}$ and $y \in \mathcal{A}$, the function $s_{y|x} : \mathbb{R}^D \to \mathbb{R}$ is continuously differentiable.*

---

[1] Modern language models typically consider the smooth Sigmoid Linear Unit (SiLU) function as an activation function, instead of, say, ReLU.

## 3.2 Variants of the loss function $\ell$

**Bradley-Terry (BT).** In DPO, and many other comparison-based preference learning models, the probability that $y$ is preferred to $z$ is then given by the classical model of Bradley & Terry (1952):

$$\mathbb{P}\left[c = 1 | x, y, z, \theta\right] \triangleq \text{SIGMOID}\left(s_{yz|x}(\theta)\right), \quad \mathbb{P}\left[c = -1 | x, y, z, \theta\right] \triangleq \text{SIGMOID}\left(-s_{yz|x}(\theta)\right),$$

where $\text{SIGMOID}(t) \triangleq 1/(1 + e^{-t})$ is the sigmoid function and $s_{yz|x}(\theta) \triangleq s_{y|x}(\theta) - s_{z|x}(\theta)$ is the score difference between responses $y$ and $z$. Assuming that the prompts and answers $x$, $y$ and $z$ are independent from $\theta$, the negative log-likelihood then defines the following loss

$$\ell(s, c) = -\log \text{SIGMOID}(cs).$$

Note that minimizing the above loss for the simplest dataset $\mathbf{D} = (x, y, z, 1)$, amounts to maximizing $\text{SIGMOID}(s)$. Since the sigmoid function is increasing, this corresponds to high values of $s$. In the DPO setting, one recovers that this favors increasing $\pi_\theta(y|x)$ and decreasing $\pi_\theta(z|x)$.

**Generalized Bradley-Terry.** The DPO and Bradley-Terry models handle "binary" comparisons, namely $c = 1$ or $c = -1$. In many situations though, one can say whether $y$ is preferable to $z$, but also by *how much*. Fageot et al. (2024) proposed a family of Generalized Bradley-Terry (GBT) models, that allow including quantified comparisons $c \in \mathcal{C}$, where $\mathcal{C} \subset \mathbb{R}$ is symmetric with respect to 0; typically, $\mathcal{C} = [-1, 1]$ or $\mathcal{C} = \mathbb{R}$. Given a score difference $s_{yz|x}$, a GBT model induces the following distribution of comparisons $c$:

$$\mathbf{p}\left[c | x, y, z, \theta\right] \triangleq \frac{f(c) \exp\left(c s_{yz|x}(\theta)\right)}{\int_{\mathcal{C}} f(\gamma) \exp\left(\gamma s_{yz|x}(\theta)\right) d\gamma},$$

where $f$ is a "root law" distribution over $\mathcal{C}$ that characterizes the GBT model. Note that the classical Bradley-Terry model is recovered by setting $\mathcal{C} = \{-1, +1\}$ and $f = (\delta_{-1} + \delta_1)/2$, where $\delta_p$ denotes the Dirac distribution at $p$. From this we can derive the loss $\ell(s_{yz|x}(\theta), c) \triangleq -\log \mathbf{p}\left[c | x, y, z, \theta\right] + cst$ as the negative log-likelihood of the data (up to a constant), we obtain

$$\ell(s, c) = \Phi_f(s) - cs,$$

where $\Phi_f(s) = \log \int_{\mathcal{C}} e^{s\gamma} f(\gamma) d\gamma$ is the cumulant-generating function of the root law $f$.

**Uniform-GBT.** For $\mathcal{C} = [-1, 1]$ and $f^{\text{unif}} = 1_{[1,1]}/2$, the loss is $\ell(s, c) = \log \frac{\sinh(s)}{s} - cs$.

**Gaussian-GBT.** For $\mathcal{C} = \mathbb{R}$ and $f(c) = \exp(-c^2/2)$, which corresponds to a normally distributed root law, the loss is $\ell(s, c) = \frac{1}{2} s^2 - cs = \frac{1}{2}(s - c)^2 - \frac{1}{2} c^2$. Up to a multiplicative rescaling of the scores, this corresponds to the variant of DPO introduced by Whitfill & Slocum (2025), where $c$ is obtained through a willingness-to-pay mechanism. We refer to Fageot et al. (2024) for a table of values of $\Phi_f$ for different root laws $f$.

**GPO losses.** Our formulation also generalizes General Preference Optimization (GPO), which proposes numerous other expressions for the loss $\ell$ (Tang et al., 2024). As they consider only binary comparisons, their loss is defined by a function $\ell_0$, such that $\ell(s, 1) = \ell_0(s)$, and $\ell(s, -1) = \ell_0(-s)$. Various expressions for $\ell_0$ are considered: $\ell_0 = -\log \text{SIGMOID}$ recovers DPO, $\ell_0(s) = \max(0, 1-s)$ recovers SLiC (Zhao et al., 2023), and $\ell_0(s) = (1 - s)^2$ recovers IPO (Azar et al., 2024). Lu et al. (2024) automatically searched and found more examples.

## 4 Pairwise Monotonicity

In this section, we introduce the notion of *pairwise monotonicity*, and prove that all models that minimize losses of our general framework are *locally pairwise monotone*.

### 4.1 Defining monotonicity

Intuitively, monotonicity holds if, whenever a preference for response $y$ over $z$ is reported, the model trained with this preference will improve the scoring of $y$ over $z$. However, precisely formulating this intuition raises a few issues.

First, different statistics of the language models may be tracked to evaluate monotonocity. Pal et al. (2024); Razin et al. (2024) considered the probability $\pi_\theta(y|x)$ of generating the preferred response given $x$. This may be called *individual-probability monotonicity*. One could also be interested to look at the individual score variations: increase of $s_{y|x}(\theta)$ and decrease of $s_{z|x}(\theta)$. We may call this criterion *individual-score monotonicity*. We discuss these notions later on, in sections 5.1 and 5.2, and show that they do not hold in general. In this section, we rather focus on the difference of scores $s_{yz|x}(\theta) = s_{y|x}(\theta) - s_{z|x}(\theta)$ between the responses $y$ and $z$. We call this *pairwise monotonicity*. Assuming that scores are the logits of the generation probabilities, pairwise monotonicity then implies a monotonicity of probability ratios, as

$$s_{yz|x}(\theta^{(2)}) \geq s_{yz|x}(\theta^{(1)}) \iff \frac{\pi_{\theta^{(2)}}(y|x)}{\pi_{\theta^{(2)}}(z|x)} \geq \frac{\pi_{\theta^{(1)}}(y|x)}{\pi_{\theta^{(1)}}(z|x)}.$$

Second, monotonicity may be measured either relative to the addition of an unequivocal comparison, or relative to an intensification of a comparison. We discuss the former in Section 4.2, and the latter in Section 4.3.

Third, in the general case, it is unclear what it means for a language model to learn from the addition of a new comparison in its dataset, or the update to an existing one, especially so if the loss function has multiple minima. To mitigate this concern, we focus on infinitesimal deviations from a critical points i.e., points which cancel the gradient; this includes minimizers. In particular, we only consider infinitesimal updates to the dataset, which yields what we call *local* monotonicity. This scenario is arguably not far from practice, given the number of data points used for training these models.

## 4.2   Pairwise monotonocity when adding an unequivocal comparison

In this section, we assume that $\mathcal{C}$ is bounded, hence has a maximum. This typically includes the settings where $\mathcal{C}$ is finite like Bradley-Terry, DPO and GPO, as well as GBT with a uniform root law on an interval or on a finite set, among many others possibilities. We then consider adding a small-weight data to $\mathbf{D}$, by defining $\mathbf{D}' \triangleq \mathbf{D} \cup \varepsilon\{(x, y, z, \max\mathcal{C})\}$, where $\mathbf{D}'$ now has $N + 1$ data, the last of which being $(x, y, z, \max\mathcal{C})$ with a weight $\varepsilon$ when it appears in Loss. Formally,

$$\text{Loss}(\theta|\mathbf{D}') \triangleq \text{Loss}(\theta|\mathbf{D}) + \varepsilon\ell(s_{yz|x}(\theta), \max\mathcal{C}).$$

**Definition 1.** *A loss* Loss *is* locally pairwise monotone *at dataset* $\mathbf{D}$ *and parameter* $\theta^*$ *for the addition of the unequivocal comparison* $(x, y, z, \max\mathcal{C})$*, if there exists a neighborhood $\mathcal{U}$ of $\theta^*$ and $\varepsilon_0 > 0$ such that, for all $x, y, z \in \mathbf{A}^*$ and for all $0 \leq \varepsilon \leq \varepsilon_0$,*

$$\forall\theta^\varepsilon \in \underset{\theta\in\mathcal{U}}{\arg\min}\,\text{Loss}(\theta|\mathbf{D} \cup \varepsilon\{(x, y, z, \max\mathcal{C})\}), \ s_{yz|x}(\theta^\varepsilon) \geq s_{yz|x}(\theta^*)$$

Intuitively, for local pairwise monotonicity to hold, a maximal comparison must push for larger score differences between $y$ and $z$. Formally, this amounts to the following.

**Assumption 2.** *The loss* $\ell : \mathbb{R} \times \mathcal{C} \to \mathbb{R}$ *is twice continuously differentiable in its first variable, and so is the regularization* $\mathbb{R}$*. Moreover, the set $\mathcal{C}$ has a maximum and $\partial_s\ell(s, \max\mathcal{C}) < 0$ for all $s \in \mathbb{R}$.*

Some versions of GPO do not verify Assumption 2, in particular for SLiC (not twice continuously differentiable) and for IPO (where saying that $y$ is preferred over $z$ pulls the score difference towards 1, even if the score difference would otherwise be larger than 1). However, the assumption holds for the classical Bradley-Terry model, and more generally, for all generalized Bradley-Terry models with a maximal comparison.

**Proposition 1.** *Assume that $\mathcal{C}$ has a maximum and that $\ell$ is derived from the Generalized Bradley-Terry model: there exists a root law $f : \mathcal{C} \to \mathbb{R}_{\geq 0}$ such that $\ell(s, c) = \Phi_f(s) - cs$. Then $\partial_s\ell(s, \max\mathcal{C}) < 0$ for all $s \in \mathbb{R}$.*

*Proof.* The GBT model with root law $f$ has loss $\ell(s, c) = \Phi_f(s) - cs$, hence $\partial_s\ell(s, \max\mathcal{C}) = \Phi_f'(s) - \max\mathcal{C}$. The derivative of the cumulant generative function is a strictly increasing odd bijection from $\mathbb{R}$ to $(\min\mathcal{C}, \max\mathcal{C})$ (Fageot et al., 2024, Theorem 1). Hence, $\Phi_f'(s) - \max\mathcal{C} < 0$. $\square$

**Theorem 1.** *Consider a preference learning model that meets Assumptions 1 and 2, a dataset $\mathbf{D}$, a data point $(x, y, z) \in \mathcal{B} \times \mathcal{A} \times \mathcal{A}$, and a parameter $\theta^\star$ such that $\nabla \text{Loss}(\theta^\star | \mathbf{D}) = 0$, and $H \triangleq \nabla^2 \text{Loss}(\theta^\star | \mathbf{D})$ is invertible. Then, $\text{Loss}$ is locally pairwise monotone at $\mathbf{D}$ and $\theta^\star$ for the addition of the unequivocal comparison $(x, y, z, \max \mathcal{C})$ if, and only if, the vector $u = H^{-1} \nabla_\theta s_{yz|x}(\theta^\star)$ is a direction of nonnegative curvature: $u^T H u \geq 0$.*

*Proof sketch.* The proof leverages the implicit function theorem, applied to the equality $\nabla \text{Loss}(\theta^\varepsilon | \mathbf{D}^\varepsilon) = 0$, which implies

$$s_{yz|x}(\theta^\varepsilon) - s_{yz|x}(\theta^\star) = -\varepsilon \partial_s \ell(s_{yz|x}(\theta^\star), \max \mathcal{C}) \nabla_\theta s_{yz|x}^T \left[ \nabla^2 \text{Loss}(\theta^\star | \mathbf{D}) \right]^{-1} \nabla_\theta s_{yz|x} + o(\varepsilon).$$

A sign analysis then allows to conclude. The full proof is given in Appendix A. □

In particular, this form of monotonicity is always satisfied at strict local optima. Moreover, Theorem 1 has an interesting consequence: at a saddle point, any failure of monotonicity yields a direction along which the loss can be minimized further (negative curvature).

Note also that Theorem 1 applies to many different comparison-based preference learning schemes, including the most popular setting of DPO. Indeed, DPO uses a Bradley-Terry loss, which is a particular instance of GBT, and thus verifies Assumption 2 (Proposition 1).

### 4.3 Pairwise monotonocity with respect to comparison intensification

We now consider monotonicity under comparison intensification. Namely, we fix a triple $(x, y, z) \in \mathcal{B} \times \mathcal{A} \times \mathcal{A}$. For any given comparison $(x', y', z', c') \in \mathcal{B} \times \mathcal{A} \times \mathcal{A} \times \mathcal{C}$, we define the $\varepsilon$-intensification of the comparison $c$ in favor of $y$ against $z$ under $x$ by

$$\text{PUSH}_\varepsilon^{x,y,z}(c' \mid x', y', z') \triangleq \begin{cases} \text{proj}_\mathcal{C}(c' - \varepsilon) & \text{if } (x', y', z') = (x, z, y), \\ \text{proj}_\mathcal{C}(c' + \varepsilon) & \text{if } (x', y', z') = (x, y, z), \\ c' & \text{otherwise}, \end{cases}$$

where $\text{proj}_\mathcal{C}(t) \triangleq \arg\min_{c \in \mathcal{C}} |t - c|$ is the projection on $\mathcal{C}$. Informally, any comparison between $y$ and $z$ on prompt $x$ is given a slight preference move towards $y$, while other comparisons are left unchanged. The $\varepsilon$-intensified dataset is then

$$\mathbf{D} + \Delta_{yz|x}^\varepsilon \triangleq \left\{ (x, y, z, \text{PUSH}_\varepsilon^{x,y,z}(c' \mid x', y', z')) \mid (x', y', z', c') \in \mathbf{D} \right\}.$$

**Definition 2.** *A loss $\text{Loss}$ with dataset $\mathbf{D}$ is locally pairwise monotone at a local minimum $\theta^\star$ for comparison intensification, if there exists a neighborhood $\mathcal{U}$ of $\theta^\star$ and $\varepsilon_0 > 0$ such that, for all $x, y, z \in \mathbf{A}^\star$, for all $0 < \varepsilon \leq \varepsilon_0$, we have*

$$\forall \theta^\varepsilon \in \arg\min_{\theta \in \mathcal{U}} \text{Loss}(\theta | \mathbf{D} + \Delta_{yz|x}^\varepsilon), \; s_{yz|x}(\theta^\varepsilon) \geq s_{yz|x}(\theta^\star)$$

The following assumption will help us characterize a family of locally pairwise-monotone preference learning models.

**Assumption 3.** *The set $\mathcal{C}$ is an interval of $\mathbb{R}$. Moreover, the loss $\ell : \mathbb{R} \times \mathcal{C} \to \mathbb{R}$ and the regularization $\mathcal{R} : \mathbb{R}^D \to \mathbb{R}$ are twice continuously differentiable, and $\partial_c \partial_s \ell(s, c) < 0$ for all score differences $s \in \mathbb{R}$ and all comparisons $c \in \mathcal{C}$.*

The latter assumption implies that $\partial_s \ell(s, c)$ is a decreasing function of $c$. Among all the examples we introduced in Section 3, the only cases where $\mathcal{C}$ is an interval are the GBT losses. It turns out that all these losses verify Assumption 3.

**Proposition 2.** *Any GBT loss whose root law has an interval support verifies Assumption 3. This includes, for instance, Uniform-GBT and Gaussian-GBT.*

*Proof.* For GBT, $\ell(s, c) = \Phi_f(s) - sc$, hence $\partial_c \partial_s \ell(s, c) = -1 < 0$. □

**Theorem 2.** *Consider a preference learning model that meets Assumptions 1 and 3, a dataset* $\mathbf{D}$, *a data point* $(x, y, z) \in \mathcal{B} \times \mathcal{A} \times \mathcal{A}$, *and a parameter* $\theta^\star$ *such that* $\nabla \mathrm{Loss}(\theta^*|\mathbf{D}) = 0$, *and* $H \triangleq \nabla^2 \mathrm{Loss}(\theta^*|\mathbf{D})$ *is invertible. Then,* $\mathrm{Loss}$ *with dataset* $\mathbf{D}$ *is locally pairwise monotone at* $\theta^\star$ *for the intensification of the comparison* $(x, y, z)$ *if, and only if, the vector* $u = H^{-1} \nabla_\theta s_{yz|x}(\theta^*)$ *is a direction of nonnegative curvature,* $u^T H u \geq 0$.

*Proof sketch.* The proof leverages the implicit function theorem to provide a first-order approximation of the new scores for the dataset $\mathbf{D} + \Delta_{yz|x}^\varepsilon$. The full proof is given in Appendix B. □

## 4.4 GLOBAL PAIRWISE MONOTONICITY UNDER STRONG CONVEXITY

Out of completeness, we show in this section that, under appropriate convexity assumptions, pairwise monotonicity holds beyond infinitesimal updates.

**Definition 3.** *A loss* $\mathrm{Loss}$ *is* globally pairwise monotone *if, for any dataset* $\mathbf{D}$, *any* $x, y, z \in \mathbf{A}^*$, *any intensification of comparisons* $yz|x$ *in* $\mathbf{D}$ *and any number of additions of comparisons* $(x, y, z, \max \mathcal{C})$ *yielding a modified dataset* $\mathbf{D}'$ *that favors more* $y$ *against* $z$ *under* $x$ *than* $\mathbf{D}$ *does,*

$$\forall \theta \in \arg\min \mathrm{Loss}(\cdot|\mathbf{D}), \ \forall \theta' \in \arg\min \mathrm{Loss}(\cdot|\mathbf{D}'), \ s_{yz|x}(\theta') \geq s_{yz|x}(\theta).$$

**Assumption 4.** *The loss* $\ell : \mathbb{R} \times \mathcal{C} \to \mathbb{R}$ *and the regularization* $\mathcal{R} : \mathbb{R}^D \to \mathbb{R}$ *are continuously differentiable. Moreover, for any* $c \in \mathcal{C}$, *and any* $(x, y, z) \in \mathcal{B} \times \mathcal{A} \times \mathcal{A}$, $\theta \mapsto \ell(s_{yz|x}(\theta), c)$ *is convex, while* $\mathcal{R}$ *is strongly convex on any compact set.*

Assumption 4 typically holds for $\ell$ convex and $s$ linear in $\theta$. In particular, it holds for any GBT model.

**Theorem 3.** *Suppose Assumptions 1 and 4 hold. Then, on one hand, Assumption 2 implies global pairwise monotonocity with respect to unequivocal comparisons. Meanwhile, on the other hand, Assumption 3 implies global pairwise monotonocity with respect to comparison intensification.*

*Proof sketch.* Because of strong convexity, the minimum is always unique, and can thus be written as a function $\theta^*(\mathbf{D})$. Now consider a continuous path $f : [0,1] \to \mathcal{D}$ with $f(0) = \mathbf{D}$, $f(1) = \mathbf{D}'$ and which continuously adds weights to unequivocal comparisons $yz|x$ or intensifies the comparisons $yz|x$ in favor of $y$. By the implicit function theorem, $\frac{d}{dt} \left[ s_{yz|x}(f(t)) \right] \geq 0$. Integrating from 0 to 1 yields the claim. The full proof is given in Appendix C. □

# 5 INDIVIDUAL MONOTONICITY

In section 4, we showed that favoring $y$ over $z$ implies an increase of $s_{yz|x}$, the score difference between the $y$ and $z$, for a wide class of comparison-based preference learning models. In this section, we consider the impact of favoring $y$ over $z$ on the scores of $y$ and $z$ individually. This brings us closer to the observations of fig. 1, that concerned the evolution of the score of the chosen and rejected alternatives, separately. In sections 5.1 and 5.2, we provide a necessary and sufficient condition for individual score monotonicity around a loss minimizer; we also consider individual probability monotonicity. Finally, in section 5.3 we consider individual score monotonicity when performing a gradient update: we provide a necessary and sufficient condition, and illustrate it on a small example.

## 5.1 LOCAL INDIVIDUAL-SCORE MONOTONICITY

Instead of score differences, we could be interested in the preferred alternative score, as in Fageot et al. (2024).

**Definition 4.** *A loss* $\mathrm{Loss}$ *with dataset* $\mathbf{D}$ *is* locally individual-score monotone *at a local minimum* $\theta^*$ *for comparison intensification, if there exists a neighborhood* $\mathcal{U}$ *of* $\theta^*$ *and* $\varepsilon_0 > 0$ *such that, for all* $(x, y, z) \in \mathcal{B} \times \mathcal{A} \times \mathcal{A}$, *for all* $0 < \varepsilon \leq \varepsilon_0$,

$$\forall \theta^\varepsilon \in \arg\min_{\theta \in \mathcal{U}} \mathrm{Loss}(\theta|\mathbf{D} + \Delta_{yz|x}^\varepsilon), \ s_{y|x}(\theta^\varepsilon) \geq s_{y|x}(\theta^*) \ and \ s_{z|x}(\theta^\varepsilon) \leq s_{z|x}(\theta^*).$$

Similarly to Fageot et al. (2024), we find a sufficient condition based on max-diagonal dominance.

**Definition 5.** *A symmetric matrix $M \in \mathbb{R}^{D \times D}$ is max-diagonally dominant if, for any $i \in [D]$, $M_{ii} \geq \max_{j \neq i} M_{ij}$.*

**Theorem 4.** *Under Assumption 3, If $\nabla \text{Loss}(\theta^* | \mathbf{D}) = 0$, $\nabla^2 \text{Loss}(\theta^* | \mathbf{D}) \succ 0$ and $\nabla s_{zy|x}(\theta^*) \neq 0$ for all $(x, y, z, c) \in \mathbf{D}$. Then $\text{Loss}$ with dataset $\mathbf{D}$ is locally individual-score monotone at $\theta^*$, for comparison intensification, if and only if the matrix $\left( \nabla_\theta s_{y|x}(\theta^*)^T \left[ \nabla^2 \text{Loss}(\theta^* | \mathbf{D}) \right]^{-1} \nabla_\theta s_{a|x}(\theta^*) \right)_{y, a \in \mathcal{A}}$ is max-diagonally dominant.*

*Proof sketch.* The proof, given in Appendix D, again leverages the implicit function theorem. □

Max-diagonal dominance is a demanding property, especially for large matrices; see e.g., Blanchard et al. (2025). Yet the matrix that is assumed to be max-diagonally dominant in Theorem 4 is of size $\mathcal{A} \times \mathcal{A}$. In the context of language models, $\mathcal{A}$ is the set of possible responses to a prompt, which is exponentially large in the response length. This suggests that local individual-score monotonicity is unlikely to hold for comparison-based preference learning algorithms in language models.

## 5.2 LOCAL INDIVIDUAL-PROBABILITY MONOTONICITY

In the context of language models, rather than scores, it is arguably more meaningful to focus on the monotonicity of probabilities (or, equivalently, of log-probabilities). We formalize this for local monotonicity, for any modification of the dataset $\mathbf{D}$.

**Definition 6.** *A loss $\text{Loss}$ with dataset $\mathbf{D}$ is locally individual-probability monotone at a local minimum $\theta^*$ for a modification of $\mathbf{D}$ into $\mathbf{D}^\varepsilon$, if there exists $\varepsilon_0 > 0$ such that, for all $(x, y, z) \in \mathcal{B} \times \mathcal{A} \times \mathcal{A}$, for all $0 < \varepsilon \leq \varepsilon_0$,*

$$\forall \theta^\varepsilon \in \arg\min_{\theta \in \mathcal{U}} \text{Loss}(\theta | \mathbf{D}^\varepsilon), \ \pi_{\theta^\varepsilon}(y|x) \geq \pi_{\theta^*}(y|x) \ and \ \pi_{\theta^\varepsilon}(z|x) \leq \pi_{\theta^*}(z|x).$$

We show that this monotonicity is vaguely linked to pairwise monotonicity. More precisely, it follows from a stronger version of pairwise monotonicity, which we call *fully pairwise monotonicity*.

**Definition 7.** *A loss $\text{Loss}$ with dataset $\mathbf{D}$ is fully pairwise monotone at a local minimum $\theta^*$ for a modification of $\mathbf{D}$ into $\mathbf{D}^\varepsilon$, if there exists $\varepsilon_0 > 0$ such that, for all $(x, y, z) \in \mathcal{B} \times \mathcal{A} \times \mathcal{A}$, for all $0 < \varepsilon \leq \varepsilon_0$,*

$$\forall \theta^\varepsilon \in \arg\min_{\theta \in \mathcal{U}} \text{Loss}(\theta | \mathbf{D}^\varepsilon), \forall w \in \mathcal{A}, \ s_{yw|x}(\theta^\varepsilon) \geq s_{yw|x}(\theta^*).$$

**Proposition 3.** *Assume that probabilities are softmax functions of the scores. Then, a fully-pairwise monotone $\text{Loss}$ is also individual-probability monotone.*

*Proof.* The proof follows by simplifying the terms of the fraction $\pi_\theta(y|x)$. See Appendix E. □

Individual-probability and fully pairwise monotonicity are very demanding, and seem unlikely to hold in practice, even locally, especially in the context of the language fine-tuning. Nevertheless, we prove the existence of an algorithm that does verify fully-pairwise monotonicity (and thus individual-probability monotonicity for softmax outputs on the scores).

**Proposition 4.** *GBT (with $s(\theta) = \theta$) is globally fully-pairwise monotone with respect to both unequivocal comparison addition and comparison intensification.*

*Proof.* The proof leverages properties of diagonally-dominant matrices. See Appendix F. □

## 5.3 GRADIENT DESCENT MONOTONICITY

The motivation of this work is to provide necessary conditions on the loss function, so that the learned models feature monotonicity guarantees. So far, our theory has focused on local (and global) monotonicity of the critical points, and in particular the minimizers, of $\text{Loss}$. Here, we change our perspective and provide necessary and sufficient conditions on the score function $s$ that guarantee that a monotone model remains monotone after one iteration of gradient descent. This iterative setting is

closer to the practice of fine-tuning language models; Figure 1, together with previous work, observed this kind of monotonicity failure.

Setting the stage, consider a model $\theta$, a loss LOSS with null regularizer $\mathcal{R} = 0$. Let $\theta^\varepsilon$ denote the model obtained after one gradient step relative to an unequivocal comparison at data point $(x, y, z)$ with learning rate $\varepsilon > 0$:

$$\theta^\varepsilon := \theta - \varepsilon \nabla_\theta \left[ \ell(s_{yz|x}(\theta), \max \mathcal{C}) \right]. \tag{1}$$

**Definition 8.** *A loss LOSS with $\mathcal{R} = 0$ is* pairwise gradient-descent (g.-d.) monotone *at $\theta \in \mathbb{R}^D$ with respect to an unequivocal comparison $(x, y, z) \in \mathcal{B} \times \mathcal{A} \times \mathcal{A}$, if there exists $\varepsilon_0 > 0$ such that for all $0 \leq \varepsilon \leq \varepsilon_0$, we have $s_{yz|x}(\theta^\varepsilon) \geq s_{yz|x}(\theta)$. Similarly, we define* fully-pairwise, *and* individual-score gradient descent (g.-d.) *monotonicity by replacing the last condition with, respectively, $s_{yw|x}(\theta^\varepsilon) \geq s_{yw|x}(\theta)$ for all $w$ in $\mathcal{A} \setminus \{y\}$, and $s_{y|x}(\theta^\varepsilon) \geq s_{y|x}(\theta)$.*

The following result translates the above notions of gradient descent monotonicity into equivalent necessary and sufficient conditions on the score function.

**Theorem 5.** *Consider a LOSS that meets Assumptions 1 and 2, let $\mathcal{R} = 0$, and $\theta \in \mathbb{R}^D$. Then, LOSS is pairwise g.-d. monotone at $\theta$ with respect to any unequivocal comparison, and there holds, at $\theta$ and relative to the addition of an unequivocal comparison $(x, y, z)$,*

$$\forall w \in \mathcal{A} \setminus \{z\}, \nabla s_{yw|x}(\theta)^T \nabla s_{yz|x}(\theta) \geq 0 \quad \Longleftrightarrow \text{LOSS } \textit{is fully-pairwise g.-d. monotone},$$
$$\nabla s_{yz|x}(\theta)^T \nabla s_{y|x}(\theta) \geq 0 \quad\quad\quad\quad\quad \Longleftrightarrow \text{LOSS } \textit{is individual-score g.-d. monotone}.$$

*Proof.* These follow from straightforward computations, which we provide in Appendix G. $\square$

**Experimental illustration** Theorem 5 provides a necessary and sufficient condition for LOSS to be monotone upon a gradient step (1) with a sufficiently small learning rate ($\epsilon \leq \epsilon_0$). Yet, it may be that $\epsilon_0$ is smaller than realistic learning rate regimes. We investigate this question for individual-score g.d. monotony on a toy problem where $s_{y|x}$ follows a ranknet architecture (Burges, 2010), and data are synthetic; see appendix H for details. Figure 2 reports the individual score difference $s_{y_t|x_t}(\theta_{t+1}) - s_{y_t|x_t}(\theta_t)$ as a function of $\nabla s_{y_t z_t|x_t}(\theta_t)^T \nabla s_{y_t|x_t}(\theta_t) \geq 0$ across the training. The individual-score guarantee of theorem 5 holds for 396 out of 400 training steps. Indeed, all iterations with a positive score difference show a positive inner product. In addition, note that negative score difference

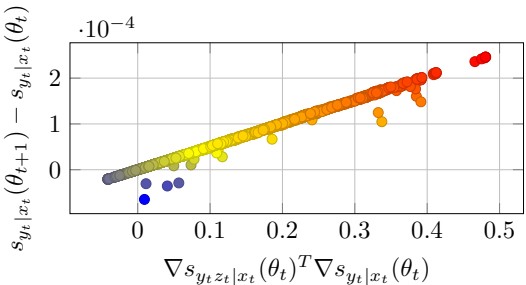

Figure 2: Scatter plot of the individual score evolution as a function of the scalar product criterion provided by theorem 5. For all but four points, the two quantities have same sign.

correspond to negative inner product, except for four points. This hints that the asymptotic development that supports the result provides a for this examples learning rate.

Theorem 5 provides a necessary and sufficient condition on the score functions so that one gradient step maintains monotonicity. These conditions provides a theoretical foundation for the design of score functions, and more generally preference learning and alignment methods, with better monotonicity properties. These conditions also apply to the online training of score models, with applications to ELO scores of chess tournament and similar contexts.

## 6 CONCLUSION

To the best of our knowledge, this paper provides the first thorough investigation of monotonicity for a very general class of comparison-based preference learning, with a focus on the effect of comparisons on the local minima, or for one gradient descent update, and through the multiple facets of monotonicity. While many previous papers pointed out deficiencies, we highlighted a noteworthy desirable property of many models, namely *local pairwise monotonicity*. We also provided insights into other forms of monotonicity.

ETHICS STATEMENT

While better improving the understanding of (non) monotonicity in preference learning, our theory does not capture other non-intuitive aspects, such as the changes of scores as shown in Figure 1. Above all, we hope to motivate more work on the mathematical guarantees of preference learning algorithms, in order to construct more trustworthy AIs (Hoang et al., 2021). Also, we caution readers against the use of preference learning algorithms from data collected in inhumane conditions, as is unfortunately mostly the case today (Höppner, 2025; Perrigo, 2023; Hao & Seetharaman, 2023; Hall & Wilmot, 2025). The very existence of data annotators in their current working conditions is one of the most pressing social issues of AI training today, it is unclear whether our work could positively contribute to this issue.

REPRODUCIBILITY STATEMENT

Code to reproduce the experiments is provided in the Supplementary Material. This includes a readme file with instructions to reproduce experiments, along with details about the hardware specifications. Details on the computing environment are also provided as the last paragraph of Section 2.

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

# Supplemental material

## A    Proofs of pairwise monotonicity for unequivocal comparisons

*Proof of Theorem 1.* Denote $\mathbf{D}^\varepsilon \triangleq \mathbf{D} \cup \varepsilon\{(x, y, z, \max \mathcal{C})\}$. We invoke the implicit function theorem for the map $\Phi : \mathbb{R}^{D+1} \to \mathbb{R}^D, (\varepsilon, \theta) \mapsto \nabla_\theta \mathrm{Loss}(\theta | \mathbf{D}^\varepsilon)$. Since $\nabla_\theta \mathrm{Loss}(\theta^* | \mathbf{D}) = 0$, we know that $\Phi(0, \theta^*) = 0$. The Jacobian matrix of $\Phi$ relative to $\theta$ is given by

$$J_\theta \Phi(\varepsilon, \theta) = \nabla^2 \mathrm{Loss}(\theta | \mathbf{D}^\varepsilon).$$

We assumed $\nabla^2 \mathrm{Loss}(\theta | \mathbf{D})$ is invertible. The implicit functions theorem thus applies, and provides the existence of $\varepsilon_0 > 0$ and a unique function $g : (-\varepsilon_0, \varepsilon_0) \to \mathbb{R}^D$ such that $g(0) = \theta^*$ and $\Phi(\varepsilon, g(\varepsilon)) = 0$ for all $\varepsilon \in (-\varepsilon_0, \varepsilon_0)$. Moreover, $g$ is differentiable and

$$g'(0) = -\left[\partial_\varepsilon J_\theta \Phi(0, \theta^*)\right]^{-1} \partial_\varepsilon \Phi(0, \theta^*) = -\left[\nabla^2 \mathrm{Loss}(\theta^* | \mathbf{D})\right]^{-1} \partial_\varepsilon \nabla \mathrm{Loss}(\theta^* | \mathbf{D}^\varepsilon)_{|\varepsilon=0}$$

Now consider any $(x, y, z) \in \mathcal{B} \times \mathcal{A} \times \mathcal{A}$, and define $\mathbf{D}^\varepsilon \triangleq$

$$\mathrm{Loss}(\theta | \mathbf{D}^\varepsilon) = \mathrm{Loss}(\theta | \mathbf{D}) + \varepsilon \ell(s_{yz|x}(\theta), \max \mathcal{C}).$$

It implies

$$\nabla_\theta \mathrm{Loss}(\theta | \mathbf{D}^\varepsilon) = \nabla_\theta \mathrm{Loss}(\theta | \mathbf{D}) + \varepsilon \partial_s \ell(s_{yz|x}(\theta), \max \mathcal{C}) \nabla_\theta s_{yz|x}(\theta).$$

Thus

$$\partial_\varepsilon \nabla_\theta \mathrm{Loss}(\theta | \mathbf{D}^\varepsilon)_{|\varepsilon=0} = \partial_s \ell(s_{yz|x}(\theta), \max \mathcal{C}) \nabla_\theta s_{yz|x}(\theta).$$

But by Assumption 2, we know that $\partial_s \ell(s_{yz|x}(\theta), \max \mathcal{C}) < 0$. In particular, we then have

$$g'(0) = \alpha \left[\nabla^2 \mathrm{Loss}(\theta^* | \mathbf{D})\right]^{-1} \nabla_\theta s_{yz|x}(\theta^*),$$

where $\alpha = -\partial_s \ell(s_{yz|x}(\theta^*), \max \mathcal{C}) > 0$. In particular, this implies that

$$
\begin{aligned}
s_{yz|x}(\theta^\varepsilon) - s_{yz|x}(\theta^*) &= s_{yz|x}(g(\varepsilon)) - s_{yz|x}(g(0)) \\
&= s_{yz|x}(g(0) + \varepsilon g'(0) + o(\varepsilon)) - s_{yz|x}(g(0)) \\
&= \nabla_\theta s_{yz|x}(\theta^*)^T g'(0)\varepsilon + o(\varepsilon) \\
&= \varepsilon \alpha \nabla_\theta s_{yz|x}(\theta^*)^T \left[\nabla^2 \mathrm{Loss}(\theta^* | \mathbf{D})\right]^{-1} \nabla_\theta s_{yz|x}(\theta^*) + o(\varepsilon), \quad (2)
\end{aligned}
$$

where we used the assumption that $s_{yz|x}$ was a differentiable function of $\theta$. Consequently, for $\epsilon$ small enough, the local pairwise monotonicity condition holds if and only if

$$u^T \cdot \nabla^2 \mathrm{Loss}(\theta^* | \mathbf{D}) \cdot u \geq 0$$

where $u = \left[\nabla^2 \mathrm{Loss}(\theta^* | \mathbf{D})\right]^{-1} \nabla_\theta s_{yz|x}(\theta^*)$. $\qquad \square$

## B    Proofs of pairwise monotonicity for comparison intensification

*Proof of Theorem 2.* The proof is very similar to the proof of Theorem 1, by now defining $\mathbf{D}^\varepsilon \triangleq \mathbf{D} + \Delta^\varepsilon_{yz|x}$ We invoke the implicit function theorem for the map $f : (\varepsilon, \theta) \mapsto \nabla_\theta \mathrm{Loss}(\theta | \mathbf{D}^\varepsilon)$, which is a function $\mathbb{R}^{1+D} \to \mathbb{R}^D$. Since $\nabla \mathrm{Loss}(\theta^*, \mathbf{D}) = 0$, we know that $f(0, \theta^*) = 0$. Note that its Jacobian matrix restricted to $\theta$ is given by

$$J_{|\theta}(\varepsilon, \theta) = \left[\partial_{\theta_j} \partial_{\theta_i} \mathrm{Loss}(\theta | \mathbf{D}^\varepsilon)\right]_{i,j \in [D]},$$

which is exactly the Hessian matrix $\nabla^2 \mathrm{Loss}(\theta | \mathbf{D}^\varepsilon)$. We assumed the Hessian to be invertible. Hence there exists $\varepsilon_0 > 0$ and a unique function $g : (-\varepsilon_0, \varepsilon_0) \to \mathbb{R}^D$ such that $g(0) = \theta^*$ and $f(\varepsilon, g(\varepsilon)) = 0$ for all $\varepsilon \in (-\varepsilon_0, \varepsilon_0)$. Moreover, $g$ is differentiable and

$$g'(0) = -\left[\partial_\varepsilon J_{|\theta}(0, \theta^*)\right]^{-1} \partial_\varepsilon f(0, \theta^*) = -\left[\nabla^2 \mathrm{Loss}(\theta^* | \mathbf{D})\right]^{-1} \partial_\varepsilon \nabla \mathrm{Loss}(\theta^* | \mathbf{D}^\varepsilon)_{|\varepsilon=0}$$

Now assume also that $(x, y, z)$ appears exactly once in $\mathbf{D}$. This can be done without loss of generality. Indeed, if it never appears, then the loss is unperturbed. If it appears multiple times, it suffices to add all the variations due to each appearance. Now, given $(x, y, z)$ appearing once in $\mathbf{D}$, we have

$$\mathrm{Loss}(\theta|\mathbf{D}^\varepsilon) = \mathrm{Loss}(\theta|\mathbf{D}) + \left(\ell(s_{yz|x}(\theta), c + \varepsilon) - \ell(s_{yz|x}(\theta), c)\right).$$

It implies

$$\nabla_\theta \mathrm{Loss}(\theta|\mathbf{D}^\varepsilon) = \nabla_\theta \mathrm{Loss}(\theta|\mathbf{D}) + \left(\partial_s \ell(s_{yz|x}(\theta), c + \varepsilon) - \partial_s \ell(s_{yz|x}(\theta), c)\right) \nabla_\theta s_{yz|x}(\theta).$$

Thus

$$\partial_\varepsilon \nabla_\theta \mathrm{Loss}(\theta|\mathbf{D}^\varepsilon)_{|\varepsilon=0} = \partial_c \partial_s \ell(s_{yz|x}(\theta), c) \nabla_\theta s_{yz|x}(\theta).$$

But by Assumption 3, we know that $\partial_c \partial_s \ell(s_{yz|x}(\theta), c) < 0$. In particular, we then have

$$g'(0) = \alpha \left[\nabla^2 \mathrm{Loss}(\theta^*|\mathbf{D})\right]^{-1} \nabla_\theta s_{yz|x}(\theta^*), \tag{3}$$

where $\alpha = -\partial_c \partial_s \ell(s_{yz|x}(\theta^*), c) > 0$ In particular, this implies that

$$s_{yz|x}(\theta^\varepsilon) - s_{yz|x}(\theta^*) = s_{yz|x}(g(\varepsilon)) - s_{yz|x}(g(0)) = s_{yz|x}(g(0) + \varepsilon g'(0) + o(\varepsilon)) - s_{yz|x}(g(0))$$

$$= \nabla_\theta s_{yz|x}(\theta^*)^T g'(0)\varepsilon + o(\varepsilon)$$

$$= \varepsilon \alpha \nabla_\theta s_{yz|x}(\theta^*)^T \left[\nabla^2 \mathrm{Loss}(\theta^*|\mathbf{D})\right]^{-1} \nabla_\theta s_{yz|x}(\theta^*) + o(\varepsilon),$$

where we used the assumption that $s_{yz|x}$ was a differentiable function of $\theta$. Consequently, for $\epsilon$ small enough, the local pairwise monotonicity condition holds if and only if

$$u^T \cdot \nabla^2 \mathrm{Loss}(\theta^*|\mathbf{D}) \cdot u \geq 0$$

where $u = \left[\nabla^2 \mathrm{Loss}(\theta^*|\mathbf{D})\right]^{-1} \nabla_\theta s_{yz|x}(\theta^*)$. $\qquad \square$

## C  GLOBAL PAIRWISE MONOTONICITY FOR CONVEX LOSS

*Proof of Theorem 3.* Make Assumptions 1, 2 and 4, and let us focus on the first part of Theorem 3. The latter part can be derived similarly.

By strong convexity of the loss (Assumption 4), not only is the minimum $\theta^*(\mathbf{D})$ unique for all datasets $\mathbf{D}$, the Hessian matrix $\nabla^2 \mathrm{Loss}(\theta^*(\mathbf{D})|\mathbf{D})$ is also guaranteed to be definite positive.

Now suppose that $\mathbf{D}'$ is obtained from $\mathbf{D}$ by $N$ operations, which are all either an addition of an unequivocal comparison to or a comparison intensification favors $y$ against $z$ under $x$. Denote $\mathbf{D}_n$ the state of $\mathbf{D}$ after the first $n$ operations. We define $f : [0, 1] \to \mathcal{D}$ as follows. For $n \in \{0, 1, \ldots, N-1\}$ and $t \in [0, 1/N)$, we define $f(n/N + t) \triangleq \mathbf{D}_n \cup (tN)\{(x, y, z, \max \mathcal{C})\}$.

By Theorem 1, we know that $s_{yz|x}(\theta^*(f(t)))$ is locally nondecreasing for all $t \in [0, 1]$. More precisely, from its proof and especially (2), we derive the fact that $s_{yz|x}(\theta^*(f(t)))$ is differentiable for all $t \in [0, 1]$ and that $\frac{d}{dt} s_{yz|x}(\theta^*(f(t))) \geq 0$ (even if $\nabla_\theta s_{yz|x}(\theta^*(f(t))) = 0$). It follows that

$$0 \leq \int_0^1 \frac{d}{dt}\left[s_{yz|x}(\theta^*(f(t)))\right] dt$$

$$= s_{yz|x}(\theta^*(f(1))) - s_{yz|x}(\theta^*(f(0)))$$

$$= s_{yz|x}(\theta^*(\mathbf{D}')) - s_{yz|x}(\theta^*(\mathbf{D})).$$

Rearranging the terms allows to conclude. $\qquad \square$

## D  PROOF OF LOCAL INDIVIDUAL SCORE MONOTONICITY

*Proof of Theorem 4.* The proof is very similar to the one of Theorem 2. Starting from (3), we have then

$$s_{z|x}(\theta^\varepsilon) - s_{z|x}(\theta^*) = s_{z|x}(g(\varepsilon)) - s_{z|x}(g(0))$$

$$= \nabla_\theta s_{z|x}(\theta^*)^T g'(0)\varepsilon + o(\varepsilon)$$

$$= \varepsilon \alpha \nabla_\theta s_{z|x}(\theta^*)^T \left[\nabla^2 \mathrm{Loss}(\theta^*|\mathbf{D})\right]^{-1} \nabla_\theta s_{zy|x}(\theta^*) + o(\varepsilon)$$

$$= \varepsilon \alpha e_z \nabla_\theta s_{|x}(\theta^*)^T \left[\nabla^2 \mathrm{Loss}(\theta^*|\mathbf{D})\right]^{-1} \nabla_\theta s_{|x}(\theta^*) e_{zy} + o(\varepsilon)$$

where the $e_z$ are elements of the canonical basis of $\mathbb{R}^D$. Finally, we have

$$s_{z|x}(\theta^\epsilon) - s_{zy|x}(\theta^*) = \beta\epsilon + o(\epsilon)$$

$$\beta_{z,y|x} \triangleq \alpha \nabla_\theta s_{z|x}(\theta^*)^T \left[\nabla^2 \text{Loss}(\theta^*|\mathbf{D})\right]^{-1} \nabla_\theta s_{zy|x}(\theta^*)$$

Therefore, individual score monotonicity is equivalent to $\beta_{z,y|x} > 0$ for all $z, y$, i.e. the max-diagonal dominance of $\nabla_\theta s_{|x}(\theta^*)^T \left[\nabla^2 \text{Loss}(\theta^*|\mathbf{D})\right]^{-1} \nabla_\theta s_{|x}(\theta^*)$.

$\square$

## E    PROOF THAT FULLY-PAIRWISE MONOTONICITY IMPLIES INDIVIDUAL-PROBABILITY MONOTONICITY

*Proof of Proposition 3.* Assuming probabilities are softmax functions of the scores, the implication follows from the fact that

$$\pi_\theta(y|x) \triangleq \frac{\exp s_{y|x}(\theta)}{\sum_w \exp s_{w|x}(\theta)} = \frac{1}{1 + \sum_{w \neq y} \exp\left(-s_{yw|x}(\theta)\right)},$$

which is an increasing function of the $s_{yw|x}$'s, for $w \in \mathcal{A}$.

Hence, $\pi_\theta(y|x)$ inherits the fully pairwise monotonicity of the scores and we have $\pi_{\theta^\epsilon}(y|x) \geq \pi_\theta(y|x)$. The proof for $z$ is similar. $\square$

## F    PROOF THAT GBT IS FULLY-PAIRWISE MONOTONE

The proof of Proposition 4 relies on the following result for diagonally dominant matrices.

**Lemma 1.** *Let $M$ be a symmetric and strictly diagonally dominant matrix (i.e. $|M_{yy}| > \sum_{z \neq y} |M_{yz}|$ for any $y$) such that $M_{yy} > 0$ and $M_{yz} \leq 0$ for any $y \neq z$. Then, its inverse $N$ satisfies*

$$N_{yy} - N_{yz} \geq N_{wy} - N_{wz} \tag{4}$$

*for any $y, z, w \in \mathcal{A}$.*

*Proof.* We first prove the following result. Assume that $a$ is a vector such that $\max_v a_v > 0$ and denote $w = \arg\max_v a_v$ so that $a_w > 0$. Then, the vector $b = Ma$ is such that $b_w > 0$. Assume by contradiction that $b_w \leq 0$. Then, we have

$$M_{ww}a_w = -\sum_v M_{wv}a_v + b_w \leq -\sum_v M_{wv}a_v.$$

However, we also have

$$\sum_v (-M_{wv})a_v \leq a_w \sum_v (-M_{wv}) < a_w M_{ww}$$

by strict diagonal dominance and using that $a_v \leq a_w$ for any $v$ and $-M_{wv} > 0$. The two inequalities are contradictory, hence $b_w > 0$.

We apply this result to $a = N_y - N_z$, the difference of the two columns $N_y$ and $N_z$ of $N$. The latter being the inverse of $M$, we have $Ma = b = e_{yz}$ where the $e_y$ are the element of the canonical basis. First, we observe that $a_y = N_{yy} - N_{yz} > 0$ due to (Fageot et al., 2024, Lemma 1). Since $y$ is the only index $w$ for which $b_w = 1 > 0$, we deduce from the previous result that $y = \arg\max_w a_w = \arg\max_w N_{wy} - N_{wz}$, which gives precisely (4). $\square$

*Proof of Proposition 4.* We can follow the proof of (Fageot et al., 2024, Theorem 2) and use Lemma 1 instead of (Fageot et al., 2024, Lemma 1) to conclude. $\square$

## G   GRADIENT DESCENT MONOTONICITY

*Proof of Theorem 5.* In this section, we assume $\mathcal{R} = 0$, and we consider the impact of sampling $(x, y, z, \max \mathcal{C})$ and of performing an infinitesimal stochastic gradient step with respect to this sample. More specifically, consider any solution $\theta \in \mathbb{R}^D$. The infinitesimal stochastic gradient step then yields

$$\theta(t + dt) = \theta(t) - \nabla_\theta \left[ \ell(s_{yz|x}(\theta), \max \mathcal{C}) \right] dt,$$

which we can rewrite

$$\frac{d\theta}{dt} = -\nabla_\theta \left[ \ell(s_{yz|x}(\theta), \max \mathcal{C}) \right] = \alpha \nabla s_{yz|x},$$

with $\alpha \triangleq -\partial_s \ell(s_{yz|x}(\theta), \max \mathcal{C}) > 0$. We then have

$$\frac{d}{dt} s_{yz|x} = \nabla s_{yz|x}^T \cdot \frac{d\theta}{dt} = \alpha \left\| \nabla s_{yz|x}(\theta) \right\|_2^2,$$

$$\frac{d}{dt} s_{y|x} = \nabla s_{y|x}^T \cdot \frac{d\theta}{dt} = \alpha \left( \nabla s_{y|x}^T(\theta) \cdot \nabla s_{yz|x}(\theta) \right),$$

$$\frac{d}{dt} s_{yw|x} = \nabla s_{yw|x}^T \cdot \frac{d\theta}{dt} = \alpha \left( \nabla s_{yw|x}^T(\theta) \cdot \nabla s_{yz|x}(\theta) \right).$$

The result follows. $\qquad\qquad\qquad\qquad\qquad\qquad\qquad\qquad\qquad\qquad\qquad\qquad\qquad\square$

## H   DETAILS ON THE INDIVIDUAL-SCORE GRADIENT DESCENT ILLUSTRATION

We provide here details on the experimental setup of the illustration of section 5.3. The code is provided as supplementary material.

We employ a dataset of $N = 50$ synthetic datapoints $(x, y, z, c)$, where $y, z \in \mathbb{R}^{10}$, $x$ is empty, and the comparison takes a random value in $\{-1, 1\}$ with uniform probability. The score function $s_{y|x}(\theta)$ is a feed-forward network with ReLU activation and one hidden layer of size $10 \times 10$. The loss function conforms to the Bradley-Terry model (see section 3.2). The optimizer is SGD with a learning rate of $10^{-3}$.

