# OpenReview forum: "On Monotonicity in AI Alignment"
_ICLR.cc/2026/Conference — Submitted to ICLR 2026_

### Official Review · Reviewer_A3S7 · 2025-10-17

**Soundness:** 3
**Presentation:** 3
**Contribution:** 2
**Rating:** 6
**Confidence:** 1

**Summary:**

This paper investigates "monotonicity violations" in AI alignment, where training a model on a preference for response 'y' over 'z' can counterintuitively cause the score of 'y' to decrease.

The main finding is a proof that while individual monotonicity can fail, a general class of methods (including DPO, GPO, and GBT) still guarantees **local pairwise monotonicity**. This means that near a stable solution, the *difference* in score between the preferred and rejected responses ($s_y - s_z$) will reliably increase.

The paper also formalizes a "bouquet" of different monotonicity types and identifies the stricter conditions required for stronger guarantees, which are often too demanding for complex language models.

**Strengths:**

1. Paper offers comprehensive theoretical take on monotonicity in post-training for LLMs on preference learning
2. Offers both weaker and stronger guarantees depending on assumptions

**Weaknesses:**

1. Exposition quite terse, some plots illustrating the intuition of local pairwise monotonicity would be helpful

**Questions:**

1. While I appreciate that this is a theory paper, what are the practical implications of this framework? Is there any insight we can use to improve the robustness during post-training?

---

> ### Author Response · Authors · 2025-11-22
>
> We thank the reviewer for their thoughtful feedback.
>
> Here is are answers to “Weaknesses & Questions”.  We have updated the manuscript based on your kind review, **changes in the updated pdf appear in blue**.
>
> > "Exposition quite terse, some plots illustrating the intuition of local pairwise monotonicity would be helpful"
>
> We thank the reviewer for raising this point.
> In order to improve clarity of exposition, we have revised the paper in the following ways.
> We updated figure 1 by swapping the curves, so that the pairs of curves to be compared are in the same plot (they were in different plots before). Let us also mention that we added one more LLM to that experiment (Qwen 0.5B), and that we updated the description for improved clarity, and precision. Observations and interpretations remain the same.
> We added a table that summarizes the main notations in the paper, located in page 2 for easy reference.
> We also added an experimental illustration to Theorem 5, showing on a small-scale example the correlation between quantities revealed by the Theorem.
>
> > "While I appreciate that this is a theory paper, what are the practical implications of this framework? Is there any insight we can use to improve the robustness during post-training?"
>
> We appreciate the opportunity to clarify this aspect.
> The practical implications of this framework are to position the notions of monotonicity in terms of (i) how relevant they are to the task of aligning the model with human preference, and (ii) the difficulty for the notion to hold in practice. Our study reveals that local pairwise monotony holds for a wide class of models (Th. 1 & 2). In contrast, fully pairwise monotonicity is too demanding for practical situations (Th. 4). This leaves the intermediate notion of gradient descent individual-score monotonicity (Sec. 5.3), which is the one relevant to the motivating example of fig. 1. There, we provide a criterion for this monotonicity to hold (Th. 5), and show its practical relevance on a small-scale problem.
> We believe that the design of alignment methods that are better behaved in terms of monotonicity is an ambitious and challenging task, outside the scope of this project, and hope that the proposed paper provides a robust step in that direction.
>
> We hope to have addressed all your concerns. We remain at your disposal may you have any further questions or require additional information.

---

> > ### Comment · Reviewer_A3S7 · 2025-11-25
> > **Response**
> >
> > Thank you for your clarifications and apologies for the delayed response. I think it sounds nice to have something to lean back on when designing new post-training objectives, so definitely opens door for further work!
> >
> > Happy to keep my score!

---

> > > ### Author Response · Authors · 2025-11-27
> > >
> > > Dear reviewer,
> > >
> > > We are glad that our answer clarified on the question you raised, and convinced you that our work is valuable for the future of designing post-trainig objectives. While we understand that evaluating LLMs and monotonicity is outside your area of expertise, is there anything we could provide that would help increase your score, or gain more confidence in the assessment?
> > >
> > > Thank you very much for your time and your work,
> > >
> > > The Authors

---

> > > > ### Comment · Reviewer_A3S7 · 2025-11-27
> > > > **Response**
> > > >
> > > > Hi,
> > > >
> > > > Thank you for your message. Tbh I can appreciate the problem formulation of inconsistent behaviour of scores when aligning LLMs, but I also have to be honest that I genuinely do not have much experience in post-training. I mainly come from a preference learning background, but when you involve policies and llm post-training formalism it becomes difficult for me to give a deeper technical evaluation since I don't have strong grasp of how these things interact, particularly in an LLM setting.
> > > >
> > > > While I appreciate the paper and the work you've done, I'm afraid I'm not the right person to confidently recommend accept or reject.

---

### Official Review · Reviewer_g8cX · 2025-10-26

**Soundness:** 2
**Presentation:** 2
**Contribution:** 2
**Rating:** 2
**Confidence:** 3

**Summary:**

The paper's goal is to study so-called "local pairwise monotonicity," in the context of RLHF: if a response y is preferred to response z, in terms of an improved score, then, will this difference or improvement (between the two responses) be increasing (monotonicity) in some given parameter (\theta); or equivalently, whether there's monotonicity in the likelihood ratio of the two scores’ corresponding probability distribution (soft max).

The paper then goes on to study the above in three specific aspects: a) the addition of an unequivocal comparison, b) relative to an intensification of a comparison, and c) infinitesimal deviations from a critical point (e.g., an optimum or zero-gradient point).

**Strengths:**

The authors apparently know what they want to study.

**Weaknesses:**

The problem is, we have no idea what's the implication of their findings in terms of helping with either SFT or inference. There are no numerical or experimental results.

**Questions:**

It seems that all four theorems are consequences of certain degrees of continuity/smoothness in an \epsilon neighborhood such that the "local pairwise monotonicity" shows. If so, then once moving out of those neighborhoods the property won't hold. Then, isn't it true that the application domain of those results is basically vacuous?

---

> ### Author Response · Authors · 2025-11-22
>
> We thank the reviewer for their feedback.
>
> We have updated the manuscript based on your kind review, **changes in the updated pdf appear in blue**.
>
> Here is are answers to “Weaknesses & Questions”.
> > "The problem is, we have no idea what's the implication of their findings in terms of helping with either SFT or inference. There are no numerical or experimental results."
>
> We agree with the review: the key contributions in our work lie in the theoretical part. However, we believe that our work still provides relevant elements for practitioners, these elements are further developed in the revised version, incorporation additional experimental illustrations, spanning different families of LLMs.
>
> We position the notions of monotonicity in terms of (i) how relevant they are to the task of aligning the model with human preference, and (ii) the difficulty for the notion to hold in practice. Our study reveals that local pairwise monotony holds for a wide class of models (Th. 1 & 2). In contrast, fully pairwise monotonicity is too demanding for practical situations (Th. 4). This leaves the intermediate notion of gradient descent individual-score monotonicity (Sec. 5.3), which is the one relevant to the motivating example of fig. 1. There, we provide a criterion for this monotonicity to hold (Th. 5), and show its practical relevance on a small-scale problem.
> We believe that the design of alignment methods that are better behaved in terms of monotonicity is an ambitious and challenging task, outside the scope of this project, and hope that the proposed paper provides a robust step in that direction.
>
> > "It seems that all four theorems are consequences of certain degrees of continuity/smoothness in an \epsilon neighborhood such that the "local pairwise monotonicity" shows. If so, then once moving out of those neighborhoods the property won't hold. Then, isn't it true that the application domain of those results is basically vacuous?"
>
> We agree with the reviewer that Th. 1 & 2 guarantee "local pairwise monotony" upon a small enough modification of the training dataset.
> Nevertheless, Th. 3 does provide a global (non-local) result that apply to practical dataset updates upon convexity assumptions.
> Such assumptions are met by simple scoring functions (eg linear one).
> As such, they do not cover LLMs, but they do cover comparison-based preference learning models such as Fageot, Farhadkhani & Hoang et al. (Generalized Bradley-Terry Models for Score Estimation from Paired Comparisons, AAAI, 2024).
> Besides, Theorem 4 proposes a local result, and yet that result is still informative. Theorem 4 provides a necessary and sufficient condition for local fully-pairwise monotonicity. That condition (max-diagonal dominance of a certain matrix) appears as too demanding to be met in practical situations, which implies that a global fully-pairwise monotonicity is a notion too demanding and unrealistic.
> Finally, we consider in Th. 5 monotonicity as one gradient step with small enough learning rate. The notion of locality there lies in the "small enough" learning rate only. While it is difficult to provide a theoretical formula for the bound, we provide a small scale experimental example, for which practical learning rates do belong to the range of small enough rates.
>
> We hope to have addressed all your concerns. We remain at your disposal may you have any further questions or require additional information.

---

> > ### Author Response · Authors · 2025-11-27
> >
> > Dear reviewer,
> >
> > As the discussion period will be coming to an end in the following days, we would appreciate your feedback on our rebuttal.
> > We believe that we have adressed all your remarks:
> > - we have added experimental results (experiments with the Qwen family, and on the score monotonicity criterion provided by Theorem 5),
> > - we have added discussions on the interpretation of our results (Th. 5),
> > - we have further commented in the rebuttal on the applicability of the results, arguing that our theorems do apply in relevant real-world situations.
> > We would be grateful if you could reconsider your score to reflect the clarified vision on our work. At the same time, we remain at your disposal should you need complementary informations.
> >
> > Thank you very much for your time and your work,
> >
> > The Authors

---

### Official Review · Reviewer_1YyV · 2025-10-31

**Soundness:** 3
**Presentation:** 2
**Contribution:** 4
**Rating:** 6
**Confidence:** 2

**Summary:**

< Summary >

This paper investigates monotonicity properties in comparison-based preference learning methods used for AI alignment.
The motivation is derived by well-known empirical observation that existing methods like DPO can counterintuitively decrease the score of the preferred response during training, despite increasing the margin between preferred and rejected responses.
To address this, the paper introduces a general framework encompassing Bradley-Terry (BT), Direct Preference Optimization (DPO), Generalized Preference Optimization (GPO), and Generalized Bradley-Terry (GBT) models and identifies what can be guaranteed and its condition. The authors formalize several flavors of monotonicity, including pairwise monotonicity, individual-score monotonicity, and individual-probability monotonicity. The findings suggest that while local pairwise monotonicity holds under mild conditions, stronger global guarantees require restrictive assumptions unlikely to hold in practice for language models.

**Strengths:**

< Strength >

- The paper addresses a practically significant issue in AI alignment. The empirical observation that preferred responses can have decreasing scores during training is widely-known concern for its reliability. The motivating example in Figure 1 effectively demonstrates this counterintuitive behavior across multiple Llama models.
- The general formulation in Section 3 successfully unifies multiple existing methods (BT, DPO, GPO, GBT) under a common loss structure and thereby enables unified study of monotonicity across a broad class of existing methods
- The mathematical rigor is generally sound, and the taxonomy of monotonicity types such as pairwise, local vs. global, score, probability, is well-organized and clarifies what different guarantees mean
- Theorem 5 provides necessary and sufficient conditions for one-step gradient descent monotonicity, which is import to actual training practice.

**Weaknesses:**

< Weakness >

- All 6 models tested are from the Llama family (3.1 8B, 3.2 3B, 3.2 1B with base/instruct variants) and this can raise concerns about generalizability. Testing on other architectures (e.g., Qwen) would strengthen confidence that findings aren't specific to Llama's particular parameterization.
- Although the paper is a theoretical analysis paper, it lacks a practical Interpretation or experiments. It would be great if the paper mention about what the theoretical guarantees mean for practitioners and show a small scale toy example. The paper doesn't clearly connect the mathematical properties to desirable empirical outcomes such as win rates or human evaluations
- The paper identifies when monotonicity fails, which is "unlikely to hold" for exponentially large response spaces). But it doesn't propose algorithmic modifications or training procedures to improve monotonicity beyond what current methods achieve. Section 5 provides sufficient conditions but no constructive way to enforce them during training.

< Minor issues >

- The paper introduces many symbols with minimal space. A notation table would improve readability.

**Questions:**

As a reviewer with a practical focus, I have a few questions

- The experiments span six models within the Llama family. To help assess the generality of the results, would you consider adding a small non-Llama baseline (e.g. Qwen-1B)? This could strengthen confidence that the findings are not family-specific.
- I would find it helpful if the authors could elaborate on the practical implications of these findings for real-world alignment applications. For instance, what should practitioners take away from knowing that local pairwise monotonicity holds? How might this inform decisions about algorithm selection, hyperparameter tuning?
- In standard DPO formulations, the “implicit reward” is often expressed via the ratio between the training and reference policies rather than directly as a logit score ​of $s_{y|x}(\theta)$. Could you clarify how score relates to the DPO implicit reward and how this mapping affects the stated monotonicity guarantees?

---

> ### Author Response · Authors · 2025-11-22
> **Official Comment by Authors (1/2)**
>
> We thank the reviewer for their thoughtful feedback.
> We are pleased that the reviewer found that our formulation successfully unifies multiple existing methods, that the proposed taxonomy of monotonicity is organized and clarifies the guarantees, and that Theorem 5 provides a relevant result to training practice.
>
> We have updated the manuscript based on your kind review, **changes in the updated pdf appear in blue**.
>
> Here is are answers to “Weaknesses & Questions”.
> > "The experiments span six models within the Llama family. To help assess the generality of the results, would you consider adding a small non-Llama baseline (e.g. Qwen-1B)? This could strengthen confidence that the findings are not family-specific."
>
> We thank the reviewer for raising this concern. As suggested, we have added one more experiment (Qwen 2.5 0.5B Base & Instruct). There also, the chosen curve is below zero for part of the training (Instruct version).
> Thus, in that case, the finetuning again effectively reduces the scores of (a subset of) the chosen alternatives during part of the training.
> Let us also mention that we updated the description of the curves of fig. 1; observations and interpretation remain the same.
>
> > "In standard DPO formulations, the “implicit reward” is often expressed via the ratio between the training and reference policies rather than directly as a logit score of $s_{y|x}(\theta)$. Could you clarify how score relates to the DPO implicit reward and how this mapping affects the stated monotonicity guarantees?"
>
> We do explain these two aspects in the submitted pdf.
> In the DPO paper, the implicit reward is defined as $r(x, y) = \beta \log \frac{\pi_r(y|x)}{\pi_{ref}(y|x)} + \beta \log Z(x)$ (their equation 5). Then, the authors model the probability that one element is preferred to an other by a Bradley Terry model, namely a binomial distribution with parameter the sigmoid of the difference of rewards (their equation (6)).
> In the submitted document, in l. 172-180, we recall the implicit reward $r(x, y)$ definition (their eq. 5) as the score $s_{y|x}$ in our notation; and in l. 188-198, we recall the Bradley Terry model, defined as a Bernoulli random variable with parameter the sigmoid of the score difference, denoted $s_{yz|x} = s_{y|x} - s_{z|x}$ in our terms.
>
> > "I would find it helpful if the authors could elaborate on the practical implications of these findings for real-world alignment applications. For instance, what should practitioners take away from knowing that local pairwise monotonicity holds? How might this inform decisions about algorithm selection, hyperparameter tuning?"
>
> > "Although the paper is a theoretical analysis paper, it lacks a practical Interpretation or experiments. It would be great if the paper mention about what the theoretical guarantees mean for practitioners and show a small scale toy example. The paper doesn't clearly connect the mathematical properties to desirable empirical outcomes such as win rates or human evaluations"
>
> We appreciate the opportunity to clarify the link to practice.
> One of our interests is to understand the reasons behind the surprising behavior of figure 1: the score of the preferred alternative decreases relative to the reference model during the finetuning of certain LLMs (eg, Llama 3.1 8B in fig. 1 among others).
> The desirable behavior is that the score of the preferred alternative does not decrease during training, or at least remain above the reference model scores.
> Our Theorem 5 provides a necessary and sufficient condition on the gradients of the score function such that individual score monotonicity holds for small enough learning rates.
> In order to investigate whether practical learning rates are covered by this Theorem, we added an additional small scale example to Section 5.3. There, it appears that the sign of the individual score difference is equal to the sign of the quantity provided by Theorem 5.
> Additionally, Theorems 1 & 2 show that pairwise monotonicity holds for models minimizing the loss, under very mild conditions.

---

> ### Author Response · Authors · 2025-11-22
> **Official Comment by Authors (2/2)**
>
> > "The paper identifies when monotonicity fails, which is "unlikely to hold" for exponentially large response spaces). But it doesn't propose algorithmic modifications or training procedures to improve monotonicity beyond what current methods achieve. Section 5 provides sufficient conditions but no constructive way to enforce them during training."
>
> We thank the reviewer for raising this point.
> We agree that designing ways to enforce individual score monotonicity during training is a crucial question. To propose a constructive way to enforce monotonicity during training appears as an ambitious research project. It is our hope that the current work will provide a robust step towards this goal, in defining (i) relevant notions of monotonicity, in showing (ii) that local pairwise monotonicity always holds locally, that (iii) fully pairwise monotonicity is too demanding, and (iv) providing a relevant condition for the relevant notion of individual monotonicity.
>
> > "The paper introduces many symbols with minimal space. A notation table would improve readability."
>
> We have added a notations table to the paper, situated in page 2.
>
> We hope to have addressed all your concerns. We remain at your disposal may you have any further questions or require additional information.

---

> > ### Author Response · Authors · 2025-11-27
> >
> > Dear reviewer,
> >
> > As the discussion period will be coming to an end in the following days, we would appreciate your feedback on our rebuttal.
> > We believe that we have adressed all your remarks:
> > - as suggested, we have added a notations table in the paper,
> > - we have added the suggested experimental results (experiments with the Qwen family), and an additional experiment on the score monotonicity criterion provided by Theorem 5,
> > - we have added discussions on the interpretation of our results (Th. 5),
> > - we have restated how DPO is captured by our framework.
> >
> > Thank you very much for your time and your work,
> >
> > The Authors

---

### Official Review · Reviewer_eWCn · 2025-11-01

**Soundness:** 4
**Presentation:** 3
**Contribution:** 3
**Rating:** 6
**Confidence:** 2

**Summary:**

A known problem in LLM fine-tuning is that pairwise monotonicity is not satisfied. This means that if in a pair $A > B$, the odds of outputting $A$ can decrease. The authors re-establish this in figures early on.

The authors define various forms of monotonicity in Sections 4 and 5, before proving a local pairwise guarantee and giving sufficient conditions for global pairwise, individual-score, individual-probability, and gradient-descent monotonicity. Section 4 in particular is where their contributions on pairswise monotonicity are detailed.

**Strengths:**

* Math is well-done; I did not make a "deep dive" but was able to follow the math and did not catch any errors or inconsistencies. A substantial piece of this paper is dedicated to theory, so this is fairly significant.

* The paper is well-motivated; the main novelty (monotonicity taxonomy and local pairwise guarantee) are clear and potential impacts with future work are clear.

* Overall, the paper is well-written. The Structure is clear and notation is consistent.

**Weaknesses:**

*The motivating example is good, but the figure is small. I would like to see a zero line, larger fonts, and clearer panel labels.

*The claim of a "toolbox" (as written in the abstract) feels somewhat strong to me. The authors do formalize several forms of monotonicity, which is appreciated, but there is not even a minimal empirical example or guidance on how to use and interpret results. I recognize space constraints but a clearer link between the sections would be appreciated.

**Questions:**

* Suggestion: as in weaknesses, I think a larger/better figure 1 would help.

* Some more anchoring in section 5 (as I stated in weaknesses) could help.

I could be persuaded to raise my score further primarily with more discussion on section 5; to me, this felt disconnected in the context of the paper. I understand why it is there, I understand why it is more "future-facing", but making the connections/flow and making it more clear and obvious the direct connection to the prior section and the value it provides in its current state would be helpful for me.

---

> ### Author Response · Authors · 2025-11-22
>
> We thank the reviewer for their thoughtful and encouraging feedback.
> We are pleased that the reviewer found the paper well-motivated, with clear novelty and potential impact, that the math is well-done, and the paper is well-written overall.
>
> We provide below a point-by-point answer to the reviewer. We have updated the manuscript based on your kind review, **changes in the updated pdf appear in blue**.
>
> > "I think a larger/better figure 1 would help"
>
> We have updated the submission with an enlarged plot. We believe that, in the new version, the zero line, fonts, and panel labels appear more clearly. We also swapped curves between rows, so that each plot now shows chosen and rejected curves of one model, thus making visual comparison easier.
>
> > "The claim of a "toolbox" (as written in the abstract) feels somewhat strong to me. The authors do formalize several forms of monotonicity, which is appreciated, but there is not even a minimal empirical example or guidance on how to use and interpret results. I recognize space constraints but a clearer link between the sections would be appreciated."
>
> We appreciate the opportunity to improve the clarity of the paper.
> We have revised the introduction, and now mention explicitly pairwise and individual monotonicity as the two topics of the paper.
> We have rephrased the "toolbox" sentence into a more specific description of the takeaways of Section 5: "Our criteria show that some flavors of individual monotonicity are too demanding for practice, and offers a criteria that predicts if one training step will maintain individual monotonicity.".
> We have also revised the discussion around our motivating example (fig. 1), highlighting how increase of margin and increase of score of chosen alternative are natural properties, to be expected during the alignment procedure on one hand, and how they are formalized by the proposed notion of pairwise (in Sect. 4) and individual monotonicity (in Sect. 5). Let us also mention that we updated the description of the curves of fig. 1; observations and interpretation remain the same.
>
> > "I could be persuaded to raise my score further primarily with more discussion on section 5; to me, this felt disconnected in the context of the paper. I understand why it is there, I understand why it is more "future-facing", but making the connections/flow and making it more clear and obvious the direct connection to the prior section and the value it provides in its current state would be helpful for me."
>
> We appreciate the opportunity to improve the flow of the paper.
> To that end, we have revised the introduction to Section 5, framing the discussion of Section 4 on Pairwise condition, and Section 5 as Individual conditions.
> We also added an illustrative example to Section 5.3, on Gradient descent monotonicity.
> We compare, on a small-scale problem, the iteration that respect individual-score monotonicity and the criterion $\nabla s_{yz|x}(\theta)^\top \nabla s_{y|x}(\theta)\ge 0$ provided in Theorem 5.
> It appears that the Theorem's claim, valid in theory only locally for small enough learning rates, actually holds for practical learning rates.
> While this does not provide directly a mean to enforce individual-score gradient monotonicity, a task beyond the scope of the current paper, it offers a promising quantity to further study.
>
> We hope to have addressed all your concerns. We remain at your disposal may you have any further questions or require additional information.

---

> > ### Comment · Reviewer_eWCn · 2025-11-27
> > **Response to Authors**
> >
> > I appreciate the authors response to my concerns, and I feel they have addressed them. I plan to monitor the conversation with other reviewers before making any decisions on altering my scores.

---

> > > ### Author Response · Authors · 2025-11-27
> > >
> > > Dear reviewer,
> > >
> > > We thank you for acknowledging we addressed all your concerns.
> > > As the end of the discussion period appraoches, and while other reviewers have not provided their feedback yet, we would be grateful if you could consider increasing your score to reflect that our responses have addressed your concerns?
> > >
> > > Thank you very much for your time and your work,
> > >
> > > The Authors

---

### Author Response · Authors · 2025-12-03
**Rebuttal Summary for AC: Reviewer Decisions and Discussion**

Dear AC and reviewers,

We sincerely thank all reviewers and the AC for their time, thoughtful, and constructive discussions. We deeply appreciate that after discussion, we have addressed almost all concerns raised by reviewers eWCn and A3S7. Reviewers 1YyV and g8cX did not get a chance to reply.

Reviewers’ Decisions After the Discussion

1. **Reviewer eWCn** offered praise for our work's quality and motivation, describing the paper as "well-motivated", "well-written", "math well-done", with "clear [...] main novelty", and "clear [...] potential impacts with future work".
The reviewer "[appreciates] the authors response to my concerns, and [feels] they have addressed them". The reviewer initial assessment is positive (6) with confidence (2), and the reviewer "[planned] to monitor the conversation with other reviewers before [...] altering scores".
2. **Reviwer 1YyV** also offered praise for our work, recognizing that our work "addresses a practically significant issue in AI alignment", which is a "widely-known concern for its reliability", and "Figure 1 effectively demonstrates this counterintuitive behavior across multiple Llama models". Our formalism "enables unified study of monotonicity across a broad class of existing methods", the "taxonomy of monotonicity types [...] is well-organized", and "Theorem 5 [studies gradient descent monotonicity, which ...] is import to actual training practice". The reviewer did not get a chance to acknowledge our rebuttal.
3. **Reviewer g8cX** initially deemed the work "reject" (2) with confidence (3), because (i) "there are no numerical or experimental results", and (ii) "it seems that [...] the application domain of these results is basically vacuous". This feedback helped us in clarifying the paper, by adding more numerical experiments, and in arguing and demonstrating numerically that our results apply in realistic situations. The reviewer did not get a chance to comment on our rebuttal.
4. **Reviewer A3S7** provided a positive assessment (6), although with low confidence (1). Our answer clarified on the reviewer's question, and convinced the reviewer that our work is valuable for the future of designing post-trainig objectives.

Summary of contributions

- **Empirical evidence of monotonicity failure**: We provide experimental evidence of failure of monotonicity during the finetuning of several LLMs (Rev. eWCn, 1YyV)
- **Pioneering general framework for alignment methods**: We propose the first theoretical framework that captures relevant cases in alignment (BT, DPO, GPO, GBT) and allows for their unified study (Rev. 1YyV, A3S7)
- **proof of local pairwise monotonicity**: We prove that local pairwise monotonicity holds under mild, realistic conditions (Rev. eWCn, 1YyV, A3S7, g8cX)
- **Sufficient condition for gradient-descent monotonicity**: we provide a sufficient condition such that one gradient step will maintain monotonicity and illustrate its relevance in practice (Rev. eWCn, 1YyV, A3S7)

Summary of clarifications and additional results from discussion

- **Improved experimental observation of monotonicity failure**: we added an experiment on Qwen model, thereby improving the confidence that monotonicity failure is not specific to Llama models, and improved plot appearance (Rev. eWCn, 1YyV).
- **Better flow of the paper**: we have reframed the abstract, added a notations table, and transition material such that Section 4 concerns Pairwise monotonicity, and Section 5 concerns Individual monotonicity (Reviewers eWCn, 1YyV, A3S7)
- **Clarified connection to practice**: we have added an illustrative example to Section 5.3 that indicates that the Theorem's claim on gradient descent monotonicity actually holds for practical learning rates (Reviewers eWCn, 1YyV, A3S7).

Regarding the locality and potential "vacuous" results raised by Reviewer g8cX

While we agree with the reviewer that our results stem from local analyses, we respectfully argue that **our results are still informative**. Indeed, Th. 3 holds globally, Th. 4 allows to conclude impossibility, and we offer numerical evidence Th. 5 covers a practical example; see the answer to Reviewr g8cX for more details.

We are pleased that both Reviewers A3S7 and eWCn, who could answer our rebuttal, have **maintained their positive scores**, with Reviewer eWCn hinting at a **potential score increase**. We hope the AC can carefully consider our discussion on the points raised by Reviewer g8cX.

We have incorporated all changes into the revised manuscript. We again thank all reviewers and the AC for their valuable time and effort.

Best regards and thanks,

Authors

---

### Meta-Review · Area_Chair_ytP4 · 2025-12-24

**Summary:**

The reviewers recognized the theoretical contributions, but all questioned the practical and algorithmic implication of the theoretical analysis.

**Reviewer Concerns:**

The authors responded to the main concern raised by all reviewers, but it is indeed not clear what this theoretical analysis suggests for the design of new algorithms. While I appreciate theoretical work that helps us better understand a phenomenom, I'm not sure why monotonicity failure is considered counter-intuitive since the BT loss is formulated not to enforce individual monotonicity, but to enforce pairwise monotonicity, which the authors prove under mild assumptions. Since the BT loss doesn't enforce individual monotonicity, it may indeed not hold in practice. For instance, when training on A>B, we want A to be more likely than B, but the likelihood of A should be allowed to decrease if both A and B are actually bad.

The authors state "Empirical evidence of monotonicity failure" as one of their contributions. However, monotonicity failure was already observed in several previous works.

**Reviewer Scores:**

I believe that all the scores would not have changed much.

---

### Decision · Program_Chairs · 2026-01-26

Reject